# Prevalence of smoking restrictions and child exposure to secondhand smoke in cars and homes: a repeated cross-sectional survey of children aged 10–11 years in Wales

Graham F Moore,[1] Laurence Moore,[2] Hannah J Littlecott,[1] Nilufar Ahmed,[1,3] Sophia Lewis,[3] Gillian Sulley,[1] Elen Jones,[1] Jo Holliday[1]

▶ Prepublication history and additional material is available. To view please visit the journal (http://dx.doi.org/10.1136/bmjopen-2014-006914).

For numbered affiliations see end of article.

**Correspondence to**
Dr Graham F Moore;
MooreG@cf.ac.uk

## ABSTRACT

**Objective:** Small increases in smoking restrictions in cars and homes were reported after legislation prohibiting smoking in public places. Few studies examine whether these changes continued in the longer term. This study examines changes in restrictions on smoking in cars and homes, and child exposure to secondhand smoke (SHS) in these locations, since 2008 postlegislation surveys in Wales.

**Setting:** State-maintained primary schools in Wales (n=75).

**Participants:** Children aged 10–11 years (year 6) completed CHETS (CHild exposure to Environmental Tobacco Smoke) Wales surveys in 2007 (n=1612) and 2008 (n=1605). A replication survey (CHETS Wales 2) was conducted in 2014, including 1601 children.

**Primary outcome variable:** Children's reports of whether smoking was allowed in their car or home and exposure to SHS in a car or home the previous day.

**Results:** The percentage of children who reported that smoking was allowed in their family vehicle fell from 18% to 9% in 2014 (OR=0.42; 95% CI 0.33 to 0.54). The percentage living in homes where smoking was allowed decreased from 37% to 26% (OR=0.30; 95% CI 0.20 to 0.43). Among children with a parent who smoked, one in five and one in two continued to report that smoking was allowed in their car and home. The percentage reporting SHS exposure in a car (OR=0.52; 95% CI 0.38 to 0.72) or home (OR=0.44; 95% CI 0.36 to 0.53) the previous day also fell. Children from poorer families remained less likely to report smoking restrictions.

**Conclusions:** Smoking in cars and homes has continued to decline. Substantial numbers of children continue to report that smoking is allowed in cars and homes, particularly children from poorer families. A growing number of countries have legislated, or plan to legislate, banning smoking in cars carrying children. Attention is needed to the impact of legislation on child health and health inequalities, and reducing smoking in homes.

## Strengths and limitations of this study

- The study reports findings from a survey of a large (n=1601) nationally representative sample of children aged 10–11 years in Wales, replicating earlier surveys in 2007/2008.
- Repeated cross-sectional surveys were conducted with the same schools in 2007/2008. More than two-thirds of those same schools were recruited in 2014. Remaining schools were replaced by schools from the same area and with comparable socioeconomic status. Samples were comparable on sociodemographic measures.
- The substantial differences in childhood reports of restrictions on smoking in cars and homes, and reports of exposure to secondhand smoke (SHS) in a car or home the previous day, between 2008 and 2014 surveys can therefore confidently be said to represent change over time.
- The study is limited by reliance on self-report measures of smoking restrictions and SHS exposure, though measures are validated against cotinine data collected in 2007/2008.
- It is not possible to make causal attributions regarding how changes over time came about.

## BACKGROUND

The dangers of secondhand smoke (SHS, or passive smoking) are now well established.[1] [2] Indeed, the WHO states that "scientific evidence has unequivocally established that exposure to tobacco smoke causes death, disease and disability".[3] Growing recognition of the dangers of SHS led many countries, including all UK countries, to implement legislation prohibiting smoking in enclosed public places and workplaces in the last decade; by 2011, an estimated 11% of the world's population lived in countries where smoking was prohibited in public spaces.[4]

In 2004, it was estimated that 61% of disease caused by SHS exposure worldwide was borne by children,[5] whose developing lungs and rapid breathing rate make them particularly vulnerable to SHS.[6] Hence, while smoke-free legislation was implemented with the primary objective of protecting adults such as hospitality workers, impacts on childhood SHS received significant international scrutiny. The case against legislation made by its opponents centred on arguments that banning smoking in public spaces would displace smoking into the home. Some evidence to support this claim was reported in Hong Kong[7] and the USA.[8] However, studies in all UK countries contradicted the displacement hypothesis. Increases in the adoption of voluntary home smoking restrictions were reported in Scotland[9][10] and England.[11] While in Wales the proportion of homes with full smoking restrictions did not change significantly,[12] fewer children reported that parents smoked inside the home after legislation.[13] Indeed, a growing body of international evidence indicates that smoke-free legislation was, in most cases, followed by increases in voluntary restrictions on smoking in private spaces.[14][15]

While the growing denormalisation of smoking around children reflected by these trends is welcome, declines in childhood SHS exposure immediately after legislation primarily benefited groups who were at relatively low risk prior to legislation. Significant declines occurred primarily among children of non-smokers[16][17] and from more affluent families.[12][13] Substantial percentages of children continued to report exposure to SHS in homes and cars. In Wales, for example, one in five children reported that smoking was allowed in their family car, while more than a third reported living in homes where smoking was allowed.[12] All measures of restrictions on smoking and childhood exposure to SHS in homes and cars indicated that, before and after legislation, exposure was particularly prevalent among children from poorer families.[12]

Debates regarding how to safeguard children from the dangers of SHS, and address the role of SHS in the intergenerational reproduction of socioeconomic inequalities, have therefore moved towards attempts to reduce smoking in cars and homes. Owing to the private nature of these spaces, regulation of behaviour is often regarded as an invasion of privacy. Hence, legislation will often only be considered where efforts to achieve change via voluntary means have not fully addressed the problem. In particular, while homes remain children's main source of SHS exposure, some have argued that only in the most authoritarian of states would legislation around smoking in the home be acceptable.[18] Hence, efforts to promote smoke-free homes remain focused on voluntary rather than legislative means.[19]

However, cars represent a space in which behaviours are already heavily regulated, hence occupying an intermediate space between public and private.[18] While children are likely to spend less time exposed to SHS inside cars than inside homes, the small and enclosed nature of vehicles means that SHS exposure is likely to be of an intense nature.[20] Furthermore, there is tentative evidence of spillover effects of banning smoking in cars, with one survey from the USA showing a substantial increase in adoption of home smoking restrictions after statewide legislation on smoking in vehicles.[21] Hence, in a growing number of countries including parts of Australia, Canada and the USA,[22] bans have been introduced on smoking in cars carrying children. Recent surveys indicate widespread public support for such a ban,[23][24] while organisations including the British Medical Association have called for a ban on smoking in all vehicles.[25] Recently, a call was issued by 600 UK respiratory health professionals for MPs to back a ban on smoking in cars carrying children.[26]

In England, a House of Commons vote in 2014 gave ministers the power to introduce a ban on smoking in cars carrying children. In Wales, the Welsh Government have attempted to restrict smoking in cars via voluntary means, announcing plans for the 'Fresh Start Wales' campaign in October 2011. This campaign, launched in 2012, comprised a range of marketing techniques through multimedia advertisements with the tagline 'Smoking in your car poisons your children', signposting to services that support quitting. The Welsh Government indicated that if insufficient voluntary changes were observed over the following 3 years, legislation would be considered, with the Children and Families Act of 2014 giving Welsh Ministers the authority to pass such legislation.

This paper presents findings of a replication of the earlier CHETS (CHild exposure to Environmental Tobacco Smoke) Wales surveys commissioned by the Welsh Government to assist with informing a decision on whether to proceed with legislation. It examines changes in children's reports of smoking restrictions and exposure to smoke in cars and homes, whether socioeconomic patterning in these variables has changed over time, and children's own attitudes towards a possible ban on smoking in cars. In summary, the paper addresses the following key research questions:

▸ Have the adoption of smoking restrictions in cars and homes increased (and children's reported exposure to SHS in these locations decreased) in Wales from 2008 to 2014?
▸ Have socioeconomic inequalities narrowed, widened or remained the same?
▸ Are increases in smoking restrictions in private spaces reported by children with parents who smoke?
▸ What are children's views on whether or not smoking in cars should be banned?

## METHODS
### Study design
CHETS Wales was a repeated cross-sectional study of year 6 (age 10–11 years) schoolchildren in 2007 and 2008. A replication study (CHETS Wales 2) was commissioned to assess changes in smoking in cars and other private spaces in 2014.

## Sampling

CHETS Wales recruited a nationally representative sample of 75 state maintained primary schools across Wales. Schools were stratified according to high/low (cut-off point identified as average entitlement across whole sample; 17.12%) free school meal entitlement (as a proxy for socioeconomic status (SES)) and Local Education Authority. Within each stratum, schools were selected on a probability proportional to school size. Where schools declined to participate, replacement schools were identified from within the same stratum. For CHETS Wales, target sample sizes were based on power to detect change in overall SHS exposure, assessed by salivary cotinine. While CHETS Wales 2 was focused on reported SHS exposure in specific locations, hence using questionnaire data, it replicated the sampling methods used for CHETS Wales. The same schools that took part in CHETS Wales were approached where possible. Schools that declined or could not be contacted were replaced with another school sampled from the same stratum. Schools were paid £50 each for their time. Within each school, one year 6 (age 10–11) class was randomly selected to participate, with all students in the class being involved.

## Consent and data collection

Consent and data collection procedures for CHETS Wales are described in detail elsewhere.[17] These were replicated for CHETS Wales 2, with the exception that no saliva samples were collected. In brief, consent was sought from schools and parents, and assent from children. Schools signed a written agreement. An opt-out consent procedure was used for parental consent in the majority of schools, with a small number requesting use of opt-in consent. Children were also assured that their participation was voluntary and given the opportunity to opt-out on the day. In all years, data were collected over a 10-week period between February and April in each year of collection. Data were collected in the classroom environment by trained staff. All staff were provided with a data collection protocol and given training in the Centre for the Development and Evaluation of Complex Interventions for Public Health Improvement (DECIPHer) to maximise standardisation of data collection procedures across the schools and data collection sweeps. Class teachers were asked to be present for disciplinary purposes, but not to intervene in the data collection in any other way unless asked to do so by the member of the research team.

## Variables

### Smoking in cars and the home

Children were asked "Is smoking allowed in your family car, van or truck?" ('yes', 'no', 'I don't know' or 'don't have a family car, van or truck') as well as "While you were inside a car yesterday was anyone smoking there?". Home smoking restrictions were assessed by asking children "Is smoking allowed inside your home?" ('No, smoking is not allowed at all', 'smoking is allowed in certain areas only', 'smoking is allowed anywhere in our home', 'smoking is allowed only on special occasions in our home', 'I don't know'). Children were also asked "While you were inside your home yesterday was anyone smoking there?". Parental smoking in the home was assessed with the question "Do any of the following people smoke in the home?" in relation to (1) father, (2) mother, (3) stepfather (or mother's partner) and (4) stepmother (or father's partner) with response options 'smokes every day', 'smokes sometimes', 'does not smoke', 'I don't know', 'I don't have or see this person'. The parent was classified as smoking in the home if the child responded 'smokes every day' or 'smokes sometimes'. Children were categorised as having (1) no parent figures who smoke in the home, (2) a father figure only who smokes in the home, (3) a mother figure only who smokes in the home and (5) two parent figures who smoke in the home.

### Objectively measured SHS exposure

Salivary cotinine (a metabolite of nicotine) is a well-validated biomarker of SHS exposure in the previous 72 h.[27] Anonymous samples were assayed using capillary gas chromatography with a detection limit of 0.1 ng/mL. Saliva samples were collected in 2007 and 2008, but not 2014. Hence, they are used to indicate the validity of self-reports of smoking in cars and homes.

### Attitudes to banning smoking in cars

In 2014, children's attitude to banning smoking in cars were assessed by asking children to circle (on a scale of 1–5) how much they agreed or disagreed with the following statements: 'There should be a complete ban on smoking in cars'; 'Smoking should be banned in cars carrying children under 16'.

### Child smoking behaviour

Respondent smoking behaviour was measured using the Office for National Statistics scale.[28] Students who gave a response other than 'I do not smoke' were classified as smokers. Additional options were 'every day', 'at least once a week' or 'less than once a week'.

### Socioeconomic status

Children completed the Family Affluence Scale (FAS[29]), which generates a composite scale based on responses to questions on bedroom occupancy, car and computer ownership, and holidays. Items were summed to form a total FAS score.

### Age

Children were asked to indicate the year and month of their birth on the smoking questionnaire. The month that the questionnaire was completed was recorded, and children's age in years calculated.

## Statistical analysis

Descriptive statistics are presented to examine the comparability of samples at 2007, 2008 and 2014 in terms of sex, age, SES, family structure and child smoking status. Significance of difference between survey years is tested using design-adjusted $\chi^2$ analyses for categorical variables and t tests for age. For all key variables other than parental smoking in the home (6.0%), data were missing in less than 5% of cases. The validity of self-report items used to assess smoking in cars and homes was examined by presenting median and IQR cotinine values, as well as the percentage of children whose saliva samples contained detectable traces of cotinine, by reported exposure. Subsequently, frequencies and percentages of children who reported exposure to SHS in cars and homes were calculated for all three time points. Significance of change from 2008 to 2014 was evaluated using logistic regression models adjusted for age and family affluence, with the year of data collection entered as the primary independent variable. ORs represent the odds of a child reporting exposure to SHS in the location specified in 2014 relative to 2008. To account for the clustered nature of the data sample, random terms for school were included in all models. These analyses were run twice: first with the entire sample, and second limited to children with at least one smoking parent. The above models were also used to examine socioeconomic inequality in smoke exposure in private spaces, through inclusion of FAS scores in the models, and testing of FAS by survey year interactions. For consistency with earlier analyses of CHETS Wales data, models including family affluence terms were limited to children living with one or both parent figures, although sensitivity analyses indicated that models which did or did not exclude children in other living arrangements gave consistent results. As a further sensitivity analysis, regression models examining change from 2008 to 2014 were re-run using only the 51 schools that took part in both years. As these produced comparable results, we report only the models using the full sample.

## RESULTS
### Response rates

Response rates for CHETS Wales are reported in detail elsewhere. In brief, 75 of 119 schools approached participated (63.0%) at both time points, with child level response rates of 91.5% and 90.4%, respectively. Of the 75 schools that participated in CHETS Wales, 4 could not be invited to participate in CHETS Wales 2 due to closure or change in status (ie, no longer a mainstream school). Of the remaining schools, 51 participated. Forty-three further schools were invited to participate before the target of 75 schools was reached (overall response rate=65.8%). Of 1862 pupils within selected classes, completed questionnaires were obtained from 1601 (86.0%). In schools where opt-out consent procedures were followed (n=74 schools, 1810 pupils), 56

children were opted-out by parents, 35 children refused and 141 were absent on the day of collection. Data were obtained from 1578 pupils (87.2%). One school requested opt-in consent. Of the 52 eligible pupils in this school, consent was given for 23 children (44.2%), all of whom provided data.

### Sample description

Pupil demographics at each time point are presented in table 1. There were no significant differences between time points, with the exception of FAS scores, which were highest in 2014. However, this was explained entirely by widespread computer ownership in 2014, with FAS scores almost identical at all time points where this item was removed. Hence, for analyses using FAS, this item is removed. FAS scores with or without computers were highly correlated (r=0.87). There were also no significant demographic differences between children within schools that participated at all time points and children within schools that did not participate again in 2014 (compared using 2008 data) or replacement schools (compared using 2014 data).

### Validity of self-reported measures of smoking restrictions and SHS exposure

Median and IQR salivary cotinine values (using 2007–2008 data), broken down by responses to self-report measures of smoking restrictions and SHS exposure, are presented in table 2. In addition, percentages of children with cotinine above the limit of detection are presented. In all cases, children who reported no smoking restrictions, or being exposed to SHS in homes or cars, provided samples with higher cotinine concentrations and were substantially more likely to provide samples containing a detectable level of cotinine. Where limited to children who reported that smoking was allowed in their home, median cotinine concentrations were seven times higher where children reported that smoking was also allowed in their car by comparison to those who said it was not (1.3 vs 0.2 ng/mL), and twice as high for children who reported being in a car where someone was smoking the previous day versus those who did not (1.6 vs 0.8 ng/mL). Hence, items on smoking in cars reflected differences in objectively measured SHS exposure which were not explained by the fact that most children who reported exposure to SHS in cars were also exposed to SHS in the home.

### Changes in smoking restrictions and self-reported exposure to SHS in cars and homes

Table 3 indicates that restrictions on smoking in cars have increased substantially since 2008, with small increases between 2007 and 2008, and more rapid changes since. For example, in 2014, 9% of children (11% of those who reported that their family own a vehicle and that they know whether or not smoking is allowed in it) reported that smoking was allowed in it, a decline from 18% (23%) in 2008. Similar declines were

**Table 1** Sample descriptions by survey year

| | Survey year | | | p Values for tests of difference | | |
|---|---|---|---|---|---|---|
| | 2007 (n=1612) | 2008 (n=1605) | 2014 (n=1601) | Comparison between years | Schools who did versus did not participate in 2014 (2008 data) | Original versus replacement schools (2014 data) |
| Boys | 778 (48.5) | 792 (49.4) | 797 (49.8) | 0.80 | 0.53 | 0.75 |
| Mean (SD) age | 10.9 (0.4) | 10.9 (0.4) | 10.9 (0.4) | 0.42 | 0.71 | 0.54 |
| Mean (SD) FAS score | 5.6 (1.9) | 5.7 (1.9) | 6.6 (1.9) | <0.001 | 0.43 | 0.90 |
| Mean (SD) FAS score without computers | 3.9 (1.5) | 3.9 (1.4) | 3.9 (1.4) | 0.41 | 0.93 | 0.71 |
| Two parent families | 1120 (69.5) | 1089 (67.9) | 1074 (67.1) | 0.37 | 0.12 | 0.57 |
| Step families | 170 (10.6) | 175 (10.9) | 152 (9.4) | | | |
| Single mother | 263 (16.3) | 273 (17.0) | 282 (17.6) | | | |
| Single father | 18 (1.1) | 23 (1.4) | 32 (2.0) | | | |
| Self-reported smokers | 24 (1.5) | 18 (1.1) | 12 (0.8) | 0.19 | 0.28 | 0.30 |

Figures are frequencies (and percentages) unless otherwise indicated.
p values for design adjusted $\chi^2$ analyses, except for age (t test).
FAS, Family Affluence Scale.

**Table 2** Salivary cotinine concentrations by responses to self-report items on exposure to secondhand smoke in cars and homes

| | | Median (and IQR) salivary cotinine concentration (ng/mL) | Frequency and percentage cotinine above limit of detection | p Value |
|---|---|---|---|---|
| Smoking allowed in car | No (n=1689) | <0.1 (<0.1 to 0.2) | 594 (35.2) | |
| | Yes (n=569) | 1.1 (0.4 to 2.2) | 526 (92.4) | <0.001 |
| | Don't know (n=424) | 0.1 (<0.1 to 0.8) | 235 (55.4) | |
| | Don't own a car (n=211) | 1.1 (0.2 to 2.7) | 179 (84.8) | |
| In a car where someone was smoking yesterday | No (n=2653) | <0.1 (<0.1 to 0.6) | 1320 (49.8) | |
| | Yes (n=196) | 1.4 (0.7 to 2.9) | 186 (94.9) | <0.001 |
| Parent figures smoke in the home | None (n=1781) | <0.1 (<0.1 to 0.1) | 588 (33.0) | |
| | Father (n=272) | 0.5 (0.1 to 1.2) | 225 (82.7) | |
| | Mother (n=299) | 1.2 (0.4 to 2.2) | 274 (91.6) | <0.001 |
| | Both (n=406) | 1.8 (1.0 to 3.0) | 396 (96.3) | |
| Smoking restrictions in the home | Full (n=1557) | <0.1 (<0.1 to 0.1) | 484 (31.1) | |
| | Partial (n=672) | 0.5 (0.1 to 1.6) | 534 (79.5) | <0.001 |
| | None (n=337) | 1.7 (0.9 to 2.9) | 319 (94.7) | |

p Values from design-adjusted $\chi^2$ analyses.

observed among children of smokers, though one in five continued to report that smoking was allowed in their family vehicle. In 2014, 4% of all children and 7% of children of smokers reported having been in a car where someone was smoking the previous day, a halving of exposure since 2008.

As indicated in table 4, percentages of children living in 'smoke-free' homes (ie, homes where smoking is not allowed at all) increased slightly between 2007 and 2008, though more markedly between 2008 and 2014. Similar changes were observed for children of smokers, among whom, half reported living in a smoke-free home in 2014, compared with 1 in 3 in 2008, while 1 in 11 lived in a home with no smoking restrictions, compared with 1 in 4 in 2008. Table 4 also indicates small declines in percentages of children reporting that one or more parent figures smoked, falling from 47% in 2007 to 40% in 2014. Larger declines were observed in percentages reporting that one or more parent figures smoked in the home. Figures for children with a parent who smoked indicate substantial reductions in the proportion of children of smokers whose parents smoked in the home, falling from 74% in 2007 to 71% in 2008 and to 52% in 2014. Hence, by 2014, almost half of children who reported that at least one parent figure smoked reported that those parent figures did not smoke in the home. The percentage of children reporting that someone was smoking in their home the previous day while they were present fell only slightly from 20.7% (n=328) in 2007 to 19.8% (n=313) in 2008 and halved to 9.6% (n=148) in 2014.

Table 5 presents ORs and 95% CIs from logistic regression models, examining change over time from 2008 to 2014 in the variables described in tables 3 and 4, and associations of SES (FAS score) with smoking in private spaces. These analyses show that all markers of exposure to SHS in cars and homes decreased significantly from 2008 to 2014. These results were maintained when the sample was restricted to those children with at least one parent figure who smokes. The likelihood of a child reporting exposure to SHS was significantly lower

for children from more affluent families in relation to all measures of exposure. There were no significant interactions between SES and survey year, with the exception of the percentage of children reporting being in a car the previous day where someone was smoking, for which socioeconomic inequalities narrowed significantly. For all remaining measures of SHS exposure, there were no significant reductions or increases in inequality.

### Children's views on smoking in cars in 2014

Among the whole sample, 71.2% (n=1109) of children agreed that smoking should be banned in cars, with 76.4% (n=1191) agreeing that smoking should be banned in cars if children were present. Where limited to children who reported that smoking was allowed in their family vehicle, a small majority agreed that smoking should be banned in all cars (55.4%; n=77) while a larger majority (61.9%; n=86) agreed that smoking should be banned in cars when children are present.

### DISCUSSION

The findings presented in this paper suggest that the denormalisation of smoking in enclosed spaces where children are present observed immediately after introduction of smoke-free legislation has continued.[11] The proportion of children who report that smoking is allowed in their family car has halved, while the percentage of children living in smoke-free homes has increased from less than two in three to almost three in four. While in 2008 a clear majority of children who lived with a parent who smoked reported that smoking was allowed in their home,[12] half now report that their home is smoke free. While it is not possible to make firm causal attributions, it is possible that this represents a continuation of the effects of smoke-free legislation, and that evaluations included follow-up periods which were too short in duration to fully capture impacts. Notably, however, other countries have reported more limited long-term progress in reducing smoking in cars and

**Table 3** Frequency (and percentage) of 10–11-year-old children in Wales reporting smoking restrictions in car

| | | Smoking allowed in family car? | | | | In car where someone smoking yesterday? |
|---|---|---|---|---|---|---|
| | | Yes | No | Don't know | No car | |
| Whole sample | 2007 | 327 (20.4) | 926 (57.8) | 231 (14.4) | 118 (7.4) | 107 (6.9) |
| | 2008 | 288 (18.0) | 965 (60.3) | 234 (14.6) | 114 (7.1) | 107 (6.7) |
| | 2014 | 141 (8.9) | 1140 (71.7) | 195 (12.3) | 115 (7.2) | 57 (3.6) |
| p Value* | | <0.001 | | | | <0.001 |
| Children with a parent who smokes | 2007 | 301 (38.6) | 272 (34.9) | 114 (14.6) | 92 (11.8) | 102 (13.5) |
| | 2008 | 259 (34.8) | 284 (38.2) | 123 (16.5) | 78 (10.4) | 98 (13.3) |
| | 2014 | 131 (19.6) | 371 (55.5) | 87 (13.0) | 79 (11.8) | 46 (7.0) |
| p Value* | | p<0.001 | | | | p<0.001 |

*p Values from design-adjusted χ² analyses.

**Table 4** Frequency (and percentage) of 10–11-year-old children in Wales reporting that parent figures smoke and levels of smoking restrictions in the home

| | No smoking parent figure | Father smokes | Mother smokes | Both smoke | p Value |
|---|---|---|---|---|---|
| 2007 | 825 (52.8) | 230 (14.7) | 187 (12.0) | 322 (20.6) | 0.01 |
| 2008 | 858 (55.5) | 235 (15.2) | 187 (12.1) | 267 (17.3) | |
| 2014 | 929 (60.2) | 211 (13.7) | 164 (10.6) | 240 (15.5) | |
| | No parent figure smokes in home | Father smokes in home | Mother smokes in home | Both smoke in home | |
| All children | | | | | |
| 2007 | 973 (63.2) | 148 (9.6) | 161 (10.5) | 258 (16.8) | <0.001 |
| 2008 | 1009 (66.8) | 144 (9.5) | 164 (10.9) | 194 (12.8) | |
| 2014 | 1153 (78.0) | 93 (6.3) | 91 (6.2) | 141 (9.5) | |
| Children with one or more parents who smoke | | | | | |
| 2007 | 192 (25.7) | 142 (19.0) | 158 (21.2) | 254 (34.1) | <0.001 |
| 2008 | 201 (29.2) | 138 (20.1) | 159 (23.1) | 190 (27.6) | |
| 2014 | 289 (47.7) | 92 (15.2) | 88 (14.5) | 137 (22.6) | |
| | Smoking in the home | | | | |
| | Full restriction | Partial restriction | No restriction | | |
| All children | | | | | |
| 2007 | 841 (59.1) | 385 (27.1) | 196 (13.8) | | <0.001 |
| 2008 | 883 (62.7) | 361 (25.6) | 164 (11.7) | | |
| 2014 | 1041 (74.3) | 303 (21.6) | 57 (4.1) | | |
| Children with one or more parents who smoke | | | | | |
| 2007 | 220 (32.0) | 285 (41.5) | 182 (26.5) | | <0.001 |
| 2008 | 218 (33.7) | 278 (43.0) | 151 (23.3) | | |
| 2014 | 294 (51.0) | 231 (40.0) | 52 (9.0) | | |

p Values from design-adjusted $\chi^2$ analyses.

homes following smoke-free legislation; in New Zealand, for example, 23% of youth reported exposure to SHS in a car in the past week in 2012.[30]

While these trends are encouraging, a large proportion of children with a parent who smokes continue to report that smoking is allowed in their home (almost half) or family car (one in five). In light of the established harms of SHS,[1][2] these levels of smoking in cars and homes still represent a significant public health concern. Furthermore, consistent with aforementioned evidence from New Zealand,[30] adoption of smoke-free homes continues to be significantly less common among poorer families. One recent paper argues that children from lower SES families are more likely to be exposed to SHS in part due to higher rates of parental smoking, but also that less affluent parents who smoke in their homes do so in greater proximity to their children, due to the smaller size of their homes.[31] Reducing socioeconomic inequalities in children's exposure to tobacco, and to SHS, remain priorities in efforts to interrupt the intergenerational reproduction of inequality.

While efforts to promote smoking restrictions in the home continue to do so through promoting voluntary change, there is widespread support for a ban on smoking in cars, from health professionals and the public.[23–26][32] This study indicates support for such a ban from children themselves, with a large majority indicating that smoking in cars carrying children should not

be allowed. Indeed, while fewer children who reported that smoking was allowed in their family car agreed with proposed legislation, a clear majority felt that smoking in cars carrying children should be banned.

Strengths of this study include its large nationally representative sample. While not all schools that took part in 2008 could be recruited again in 2014, the 2014 survey successfully recruited two-thirds of the schools that took part in the earlier CHETS Wales study, and achieved a sample with no significant demographic differences to the original sample. While we are unable to make causal attributions regarding how changes occurred, differences between survey years can be confidently considered to reflect change over time rather than sampling differences. The study relies on self-reports of SHS exposure. However, while no saliva samples were collected in 2014, for all self-reported indicators of smoking restrictions and SHS exposure in cars and homes, objective indicators were consistent with children's reports in 2007/2008. Hence, changes in self-reports of smoking restrictions and SHS exposure can be confidently assumed to reflect meaningful reductions in SHS exposure.

Partly informed by the key findings from this study, the Welsh Government announced that it will introduce legislation banning smoking in cars carrying children similar to that in place in parts of Canada, Australia and the USA,[22] citing the high proportion of children with parents who smoke who are still exposed to smoke in cars. Further research is needed to

**Table 5** ORs and 95% CIs from logistic regression models examining associations of year of data collection and SES with exposure to smoke in private spaces

| | Smoking allowed in cars (yes vs no) | Smoking in car yesterday | Smoking restriction in the home (base category=full restriction) | | Smoking in home yesterday | Parent figures smoke in the home | | |
| --- | --- | --- | --- | --- | --- | --- | --- | --- |
| | | | Partial | No restriction | | Father only | Mother only | Both parents |
| **All children** | | | | | | | | |
| n | 2407 | 2987 | 2664 | | 2955 | 2836 | | |
| **Model 1** | | | | | | | | |
| Year (reference=2008) | **0.42 (0.33 to 0.54)** | **0.52 (0.38 to 0.72)** | **0.70 (0.59 to 0.83)** | **0.30 (0.20 to 0.43)** | **0.44 (0.36 to 0.53)** | **0.54 (0.42 to 0.70)** | **0.48 (0.36 to 0.64)** | **0.65 (0.49 to 0.86)** |
| FAS | **0.74 (0.68 to 0.80)** | 0.92 (0.83 to 1.02) | **0.77 (0.72 to 0.83)** | **0.63 (0.57 to 0.71)** | **0.70 (0.65 to 0.75)** | **0.73 (0.67 to 0.81)** | **0.72 (0.65 to 0.79)** | **0.67 (0.62 to 0.73)** |
| **Model 2** | | | | | | | | |
| FAS*year | 1.14 (0.95 to 1.37) | **1.28 (1.01 to 1.60)** | 1.00 (0.88 to 1.14) | 1.05 (0.80 to 1.38) | 1.06 (0.91 to 1.22) | 0.94 (0.78 to 1.15) | 1.06 (0.84 to 1.34) | 1.05 (0.86 to 1.29) |
| **Children with at least one parent figure who smokes** | | | | | | | | |
| n | 982 | 1303 | 1149 | | 1303 | 1217 | | |
| Year (reference=2008) | **0.41 (0.31 to 0.53)** | **0.49 (0.35 to 0.69)** | **0.59 (0.48 to 0.73)** | **0.26 (0.17 to 0.39)** | **0.41 (0.33 to 0.51)** | **0.45 (0.33 to 0.60)** | **0.38 (0.28 to 0.53)** | **0.52 (0.38 to 0.70)** |
| FAS | **0.87 (0.79 to 0.97)** | 1.07 (0.96 to 1.20) | **0.82 (0.75 to 0.88)** | **0.70 (0.61 to 0.80)** | **0.80 (0.75 to 0.87)** | **0.88 (0.78 to 0.98)** | **0.86 (0.77 to 0.95)** | **0.79 (0.71 to 0.88)** |

All models adjusted for age and include random terms for school. Significant ORs highlighted in bold.
FAS, Family Affluence Scale; SES, secondhand smoke.

understand the impacts of this legislation on childhood SHS exposure (including compliance with legislation, and effects on smoking behaviour in other locations, such as the home),[21] health outcomes and health inequalities. In addition, there is a need for sustained attention to understanding how to reduce smoking in the main location in which children continue to be exposed to SHS, the home. Further reducing childhood SHS exposure, while eliminating socioeconomic inequality, will likely require a combination of efforts to help parents to successfully quit smoking, and to support those who continue to smoke in not doing so in the home.

**Author affiliations**
[1]Centre for the Development and Evaluation of Complex Interventions for Public Health Improvement (DECIPHer), School of Social Sciences, Cardiff University, Cardiff, UK
[2]MRC/CSO Social & Public Health Sciences Unit, University of Glasgow, Glasgow, UK
[3]School of Social Sciences, Cardiff University, Cardiff, UK

**Acknowledgements** The authors thank Chris Roberts and Ian Jones (Social Research and Information Division, Welsh Government) for support and advice; and the Public Health Division (Welsh Government) for funding the study, and the schools and schoolchildren who participated in the study. They also thank Natalie Richards and Kim Sheppard for administrative assistance, Sophia Lewis for assistance with questionnaire design, and all fieldworkers who assisted with data collection.

**Contributors** GFM, JH and LM were investigators on the CHETS 2 study, and were involved in study conception and design. The set-up of the survey was managed by NA, under the supervision of GFM and JH, and the conduct of the survey was managed by NA, under the supervision of GFM. GFM developed the paper plan, led data analysis and drafting of the manuscript. HJL assisted with data analysis and drafting of the manuscript. SL wrote the first draft of a literature review which informed the background section. All authors contributed to drafts of the full manuscript and approved the final draft.

**Funding** The lead author is supported by an MRC Population Health Scientist Fellowship (MR/K021400/1). The study was funded by the Public Health Division, Welsh Government. The work was undertaken with the support of The Centre for the Development and Evaluation of Complex Interventions for Public Health Improvement (DECIPHer), a UKCRC Public Health Research Centre of Excellence. Joint funding (MR/KO232331/1) from the British Heart Foundation, Cancer Research UK, Economic and Social Research Council, Medical Research Council, the Welsh Government and the Wellcome Trust, under the auspices of the UK Clinical Research Collaboration, is gratefully acknowledged.

**Competing interests** None.

**Ethics approval** Cardiff University School of Social Science Research Ethics Committee.

**Provenance and peer review** Not commissioned; externally peer reviewed.

**Data sharing statement** No additional data are available.

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
