## [Reviewer comments · BMJ Open]

ARTICLE DETAILS

TITLE (PROVISIONAL)	Prevalence of smoking restrictions and child exposure to secondhand smoke in cars and homes: a repeated cross-sectional survey of 10-11 year old children in Wales
AUTHORS	Moore, Graham; Moore, L; Littlecott, Hannah; Ahmed, Nilufar; Lewis, Sophia; Sulley, Gillian; Jones, Elen; Holliday, Jo

VERSION 1 - REVIEW

REVIEWER	Alessandra Lugo IRCCS - Istituto di Ricerche Farmacologiche "Mario Negri", Milan, Italy
REVIEW RETURNED	27-Nov-2014

GENERAL COMMENTS	In the present manuscript, authors report the prevalence of children aged 10-11 years exposed to second-hand smoke in private cars and in homes in Wales in 2014, and compare the results with a previous study conducted in 2008, a year later the implementation of a legislation prohibiting smoking in enclosed public places and workplaces. The topic is relevant and the manuscript is well-written. Comments: 1) The Discussion is mainly focused on re-reporting the results rather than on "discussing" them. I suggest to the authors to shorten that part and give more focus to, for example, the comparison with other studies investigating the same research issue.2) In Table 1 authors give the frequency and percentage of children self-reporting to be smokers. How was this variable assessed? And which were the possible answers? Please, specify this in the Variables sections of the Methods.3) Are the results provided in Table 1, 3 and 4 supported by statistical tests (t-test for continuous variables, or chi-squared for categorical ones)?4) Table 5 should be self-explicative and need to be revised, thus I suggest authors the following changes:a. provide in a footnote the model used (adjusting variables and random effects);b. report the reference categories for year and FAS (i.e., 2014 vs. 2008, high vs. low FAS);c. please, double check the second column of the "smoking restriction in the home": is "none" correct or it should be "full"?d. replace "children of smokers" with "children with at least one parent who smoke";e. change the symbol * to indicate that ORs in bold are those significant: it can be confused with the symbol of interactions between year and FAS. Minor comment: 5) Please, provide the acronym of CHETS at the first occasion.
---

REVIEWER	Sara Hitchman Department of Addictions, Institute of Psychiatry, Psychology & Neuroscience, King's College London
REVIEW RETURNED	27-Nov-2014

GENERAL COMMENTS	Overall: Important paper reporting new data. However, reporting of results is not clear (tables, no p-values, absence of any statistical tests to support statements declaring significance, declines, etc) Abstract: Clear, but distinction between restrictions in home vs. smoking in front of children should be made clear. Strengths and limitations - Typo, 20007, delete zero. - Also state it is cross-sectional in this section, and some different schools - Again, clearly distinguish what is being reported, reported exposure to SHS in cars and homes, or rules about smoking in cars and homes Introduction Missing reference for statement that children's rapid breathing and lung development make them vulnerable, the only place I have seen this written is here: Bearer CF. Environmental health hazards: How children are different from adults. Future of Children 2005; 5: 11-26. Page 6, line 3, do you mean around, not among children?? Good clearly defined research questions, again please just match introduction points to research questions where possible, refer to rules/restrictions on smoking in homes and cars and what was measured, talking about exposure implies that that exposure was measured (e.g., cotinine), the important thing is to be clear. Methods Please give age of year 6 children for international readers not familiar with Welsh school system at line 18 when first mentioned, page 8 Variables This is the first place where the distinction is made clear between the exposure measure (yesterday) and the rule/restriction measure, please make clearer earlier on. Statistical Analysis Good. But after reading sample description, some things omitted...p-values for tests? Tests of changes between 2008 – 2014? Sample description Please describe tests that were used to make the statement at page 12, line 45 'no significant demographic differences between children attending schools ...' Descriptive comparisons cannot to be used to make such a statement. Results Tables 2, 3, and 4 cut across 2 pages, this makes it hard to read. Provide test statistics for statements such as one on line 31-32, page 13, 'substantially more likely to provide samples containing cotinine...' Page 15, line 8, half of children with smoking parents still living in homes where smoking is allowed is shockingly still very high given the consequences of SHS exposure, despite the large fall over the past seven years... Maybe elaborate on this in the discussion. Children's views on smoking in cars, at page 16, line 45, be clear on year, I assume 2014. Table 3 and 4. Where are statistical tests comparing 2008 to 2014? Cannot make conclusive statements without.
--

	Table 5 – p-values? Need to include. Small n should be used within the tables. Please ensure ORs do not cut across two lines. Discussion Page 27, line 18, increased to 3 in 4 from what? Page 18, line 38, With half of parents who smoke still allowing smoking in the home, conclusion cannot really be made that not allowing smoking in the home is now the norm... Page 19, line 19, would also be helpful to restate the % support here to remind the reader There needs to be more discussion of limitations, cross-sectional, different samples of schools, etc. Overall, please review similar papers comparing changes in cross-sectional school surveys. The most recent I can easily recall is: J Adolesc Health. 2014 Nov;55(5):713-5. doi: 10.1016/j.jadohealth.2014.07.015. Rise in electronic cigarette use among adolescents in Poland. Goniewicz ML1, Gawron M2, Nadolska J3, Balwicki L3, Sobczak A4
--	---

VERSION 1 – AUTHOR RESPONSE

COMMENTS FOR AUTHORS:

In the present manuscript, authors report the prevalence of children aged 10-11 years exposed to second-hand smoke in private cars and in homes in Wales in 2014, and compare the results with a previous study conducted in 2008, a year later the implementation of a legislation prohibiting smoking in enclosed public places and workplaces. The topic is relevant and the manuscript is well-written.

- We thank the reviewer for their positive overall assessment of our manuscript. We have made the requested clarifications as indicated below.

The Discussion is mainly focused on re-reporting the results rather than on “discussing” them. I suggest to the authors to shorten that part and give more focus to, for example, the comparison with other studies investigating the same research issue.

- We have redrafted sections of the discussion to shorten reiterations of the results, and to discuss comparisons with other relevant studies.

In Table 1 authors give the frequency and percentage of children self-reporting to be smokers. How was this variable assessed? And which were the possible answers? Please, specify this in the Variables sections of the Methods.

- This has now been added to the methods section

Are the results provided in Table 1, 3 and 4 supported by statistical tests (t-test for continuous variables, or chi-squared for categorical ones)?

- We now indicate p-values for comparisons between samples in terms of socio-demographic characteristics for the 3 years, to evidence our conclusion that samples did not differ by survey year. In tables 3 and 4, we had initially presented only descriptive information for variables on smoking restrictions and exposure to SHS in cars and homes. We subsequently test the significance of changes in these variables from 2008-2014 using the regression models presented in Table 5. We now also include p-values from design-adjusted chi-squared analyses for all variables within Tables 3 and 4.

Table 5 should be self-explicative and need to be revised, thus I suggest authors the following changes:

- provide in a footnote the model used (adjusting variables and random effects);
 - This has now been added
- report the reference categories for year and FAS (i.e., 2014 vs. 2008, high vs. low FAS);

- A reference category has been added for year. As FAS was treated as a continuous variable, consistent with our earlier analyses of the CHETS Wales survey, there is no reference category. This is now clarified in the description of the measure.

c. please, double check the second column of the “smoking restriction in the home”: is “none” correct or it should be “full”?

- The label was correct, as it related to there being no restriction in place. The base category for these models was ‘full smoking restriction’. This is now clarified

d. replace “children of smokers” with “children with at least one parent who smoke”;

- This change has been made

e. change the symbol * to indicate that ORs in bold are those significant: it can be confused with the symbol of interactions between year and FAS.

- This symbol has now been removed

Minor comment:

Please, provide the acronym of CHETS at the first occasion.

- This has now been added

Reviewer: 2

Overall: Important paper reporting new data. However, reporting of results is not clear (tables, no p-values, absence of any statistical tests to support statements declaring significance, declines, etc)

- We thank the reviewer for their comments. Most relate to the reporting of our results, and we have included most of the requested additional statistics. The main additional comment which runs throughout the review is a need for greater clarity about the fact that we are reporting on measures of restrictions relating to smoking in cars and homes, rather than just exposure. Hence, we have altered the title, and edited throughout, to be clear that our measures relate both to restrictions and to self-reported exposure in cars and homes.

Abstract:

Clear, but distinction between restrictions in home vs. smoking in front of children should be made clear.

- The abstract has been edited to clarify this point.

Strengths and limitations

Typo, 20007, delete zero.

- This has been deleted

Also state it is cross-sectional in this section, and some different schools

- The repeated cross sectional study design is now described. The fact that some schools differed between survey years was indicated already.

Again, clearly distinguish what is being reported

- As with the rest of the manuscript, we have edited to indicate that we are measuring restrictions on smoking in cars and homes as well as self-reported exposure.

Introduction

Missing reference for statement that children’s rapid breathing and lung development make them vulnerable, the only place I have seen this written is here: Bearer CF. Environmental health hazards: How children are different from adults. Future of Children 2005; 5: 11-26.

- This reference has been added

Page 6, line 3, do you mean around, not among children??

- This sentence has been amended

Good clearly defined research questions, again please just match introduction points to research questions where possible, refer to rules/restrictions on smoking in homes and cars and what was measured, talking about exposure implies that that exposure was measured (e.g., cotinine), the important thing is to be clear.

- We have edited the introduction and discussion to be clear that we are examining changes in restrictions relating to smoking in cars and homes, as well as self-reported exposure in those locations.

Methods

Please give age of year 6 children for international readers not familiar with Welsh school system at line 18 when first mentioned, page 8

- This has been added

Variables

This is the first place where the distinction is made clear between the exposure measure (yesterday) and the rule/restriction measure, please make clearer earlier on.

- We have edited the introduction and discussion to be clear that we are examining changes restrictions around smoking in private spaces, as well as self-reported exposure.

Statistical Analysis

Good...But after reading sample description, some things omitted...p-values for tests? Tests of changes between 2008 – 2014?

- These points are now addressed individually below.

Sample description

Please describe tests that were used to make the statement at page 12, line 45 'no significant demographic differences between children attending schools' Descriptive comparisons cannot to be used to make such a statement.

- P-values for tests of differences between samples in demographic variables are now reported in Table 1, including comparisons by survey year, and comparisons between drop-outs/replacement schools and those who provided data at both time-points.

Results

Tables 2, 3, and 4 cut across 2 pages, this makes it hard to read.

- This has been amended

Provide test statistics for statements such as one on line 31-32, page 13, 'substantially more likely to provide samples containing cotinine....'

- P-values from design-adjusted chi-squared analyses have now been added into Table 2 to back up the statements made in this section.

Page 15, line 8, half of children with smoking parents still living in homes where smoking is allowed is shockingly still very high given the consequences of SHS exposure, despite the large fall over the past seven years... Maybe elaborate on this in the discussion.

- In our discussion, we indicate that while declines are encouraging, substantial subgroups of children continue to be exposed to SHS.

Children's views on smoking in cars, at page 16, line 45, be clear on year, I assume 2014.

- The year (2014) is now indicated in the title of this section

Table 3 and 4. Where are statistical tests comparing 2008 to 2014? Cannot make conclusive statements without.

- The significance of changes in these variables from 2008-2014 was analysed using logistic regression models, which are reported in Table 5. We now also include p-values from design-adjusted chi-squared tests within Tables 3 and 4.

Table 5 – p-values? Need to include. Small n should be used within the tables. Please ensure ORs do not cut across two lines.

- Given the large number of values within this table, we feel that addition of p-values would make the table congested and difficult to read, but indicate which ORs are significant at the 95% level through use of bold text. It is common practice when reporting odds ratios from logistic regression models not to report p-values. Significance can be ascertained by examining whether confidence intervals pass through 1.00. BMJ Open have published a number of articles which report logistic regression outcomes in this way (see for example <http://bmjopen.bmj.com/content/4/6/e005002.full.pdf+html>).

Discussion

Page 27, line 18, increased to 3 in 4 from what?

- This has now been clarified

Page 18, line 38, With half of parents who smoke still allowing smoking in the home, conclusion cannot really be made that not allowing smoking in the home is now the norm...

- We did not claim that not allowing smoking was the norm but that "it is no longer clearly the norm for smoking to be allowed in the home". However, in thinning down our reiteration of the results in response to comments from Reviewer 1, this statement has now been removed.

Page 19, line 19, would also be helpful to restate the % support here to remind the reader

- The percentage has now been indicated

There needs to be more discussion of limitations, cross-sectional, different samples of schools, etc.

- The study is a repeated cross-sectional design which examines change over time. Although we make no attempts to do so, we have indicated in discussion of strengths limitations that we are unable to make causal attributions about how change occurred. The fact that not all of the 75 schools who took part in 2008 took part again in 2014 was already acknowledged, but is now emphasised more explicitly.

Overall, please review similar papers comparing changes in cross-sectional school surveys over time.

The most recent I can easily recall is: *J Adolesc Health*. 2014 Nov;55(5):713-5. doi:

10.1016/j.jadohealth.2014.07.015. Rise in electronic cigarette use among adolescents in Poland.

Goniewicz ML1, Gawron M2, Nadolska J3, Balwicki L3, Sobczak A4

- The specific article suggested here is not included, as it focuses on child smoking and e-cigarette use, rather than the core issues we report here (i.e. smoking in cars and homes). We have however expanded on the number of references, including studies from New Zealand and the US, examining change over time in restrictions on smoking in cars and homes.